# Oral Papillomatosis: Its Relation with Human Papilloma Virus Infection and Local Immunity—An Update

**DOI:** 10.3390/medicina58081103

**Published:** 2022-08-15

**Authors:** Elena Cristina Andrei, Ileana Monica Baniță, Maria Cristina Munteanu, Cristina Jana Busuioc, Garofița Olivia Mateescu, Ramona Denise Mălin, Cătălina Gabriela Pisoschi

**Affiliations:** 1Department of Histology, Faculty of Dentistry, University of Medicine and Pharmacy, 2-4 Petru Rareș Street, 200349 Craiova, Romania; 2Department of Oro-Maxillofacial Surgery, Faculty of Dentistry, University of Medicine and Pharmacy, 200349 Craiova, Romania; 3Department of Histology, Faculty of Medicine, University of Medicine and Pharmacy, 200349 Craiova, Romania; 4Department of Otorhinolaryngology, University of Medicine and Pharmacy, 200349 Craiova, Romania; 5Department of Pharmaceutical Biochemistry, University of Medicine and Pharmacy, 200349 Craiova, Romania

**Keywords:** oral papilloma, oral HPV infection, HPV-related oral lesions, HPV immunity, HPV oncogenicity

## Abstract

Oral papilloma lesions may appear as a result of HPV infection, or not, and only special molecular methods could differentiate them. Low-risk and high-risk HPV types could induce oral HPV papillomatosis with different natural evolution, clearance and persistence mechanisms. The pathogenic mechanisms are based on the crosstalk between the oral epithelial and immune cells and this very efficient virus. HPV acts as a direct inducer in the process of transforming a benign lesion into a malignant one, the cancerization process being also debated in this paper. According to the degree of malignity, three types of papillomatous lesions can be described in the oral cavity: benign lesions, potential malign disorders and malignant lesions. The precise molecular diagnostic is important to identify the presence of various virus types and also the virus products responsible for its oncogenicity. An accurate diagnostic of oral papilloma can be established through a good knowledge of etiological and epidemiological factors, clinical examination and laboratory tests. This review intends to update the pathogenic mechanisms driving the macroscopic and histological features of oral papillomatosis having HPV infection as the main etiological factor, focusing on its interreference in the local immunity. In the absence of an accurate molecular diagnostic and knowledge of local immunological conditions, the therapeutic strategy could be difficult to decide.

## 1. Introduction

Oral papillomatosis is a heteromorphic group of benign lesions of the oral mucosa whose cause has not been fully elucidated yet. The importance of human papillomavirus (HPV) as an etiological factor of papillomatous lesions is considerably increased with the development of molecular biology techniques used for diagnostics, but papillomatosis has also a non-viral etiology, various factors producing the chronic irritation of the oral mucosa for a long time or having a detrimental effect on tissues being involved: excessive smoking, chronic alcoholism [1,2,3,4] or mucosal trauma producing chronic irritation such as incorrect prosthetic works [5] or edentations [3,4,6]. The accumulation of bacterial plaque as a result of poor oral hygiene and tartar may also cause oral papilloma lesions [7], and recently, another etiological factor incriminated is the lichen planus [8,9,10,11]. The macroscopic and histological aspects of non-viral and viral-induced oral papillomatosis being similar, a complete and reliable diagnosis can be established after performing molecular methods that accurately detect the viral presence.

The same etiological factors inducing papillomas are incriminated in the development of malignant tumors of the oral cavity, part of the head and neck squamous cell carcinoma (HNSCC) [12,13], classified as HPV-positive and HPV-negative [14,15,16,17]. To date, the HNSCC is a heterogeneous group of diseases in terms of risk factors, pathogenesis and prognostic outcome. According to the topographical criteria, the subtypes of HNSCC are located in the oral cavity, oropharynx, hypopharynx, larynx and nasopharynx [18]. HPV drives the oropharyngeal squamous cell carcinoma (OPSCC), while tobacco and alcohol consumption are responsible for other locations of HNSCC [14,15,16,17,19].

The incidence of HPV viruses varies by the geographic area, being higher in underdeveloped countries (East African countries), with the lowest incidence in West Asia [20,21]; the economic development and the efficiency of the medical system are also important: countries in which the HPV vaccination program has been successfully implemented have a significant decrease in the incidence rate [22,23].

HPVs are part of the Papillomaviridae family, non-enveloped viruses with a DNA double-stranded circular genome [24,25,26,27]. If about 70 types of HPV were recognized 10 years ago, now over 200 different genotypes have been identified based on viral genome sequences [1,14,26,28]. HPVs have tropism for the squamous epithelium and they comprise five genera, designated by Greek letters, with different mucosal or cutaneous tropism and disease association; only the alpha genus includes viruses with mucosal and cutaneous tropism, the other four causing only cutaneous lesions [29,30]. In the alpha and beta genera, the Agency for Research on Cancer (IARC) described three groups of HPVs according to their carcinogenic effects on human mucosa: (i) HPV-16 is defined as the most potent HPV type causing cancer at several sites: cervix, vulva, vagina, penis, anus, oral cavity, oropharynx, tonsils and larynx; (ii) HPV-18, 31, 33, 35, 39, 45, 51, 52, 56, 58 and 59—sufficient evidence for cervical cancer; (iii) HPV-5 and HPV-8 were included in the beta group with limited evidence for skin cancer in patients with epidermodysplasia verruciformis [31].

Based on their association with cervical cancer, HPV types are classified as high-risk (HPV-16, 18, 31, 33, 35, 39, 45, 51, 52, 56, 58, 59, 68, 73 and 82) or low-risk (HPV-26, 30, 34, 53, 66, 67, 69, 70, 73, 82 and 85) [16]. Ten of these HPVs, HPV-16, 18, 31, 33, 45, 51, 52, 56, 58 and 59, have been incriminated in HNSCC [14,32]. The oncogenic types, 16, 18, 31 and 33, are sexually transmitted [18,33]. For an exhaustive and recent classification of HPVs and their relationship with infection lesions, see also [18,33].

Low-risk types of HPV also infect the oral cavity, but the pathogeny and the evolution of the subsequent lesions are different. Frequently, low-risk strains produce warts in the oral mucosa, while high-risk strains can lead to intraepithelial squamous lesions evolving into oral carcinoma [1,34]. Low-risk HPV infection is associated with a subset of atypical papillomatous lesions with a benign clinical course mainly in immunosuppressed subjects [35]. It is mentioned that low-risk HPV types 6 and 11 determine the formation of papilloma lesions in the larynx, a pathological condition named recurrent respiratory papillomatosis (RRP) [36]. Much less is known about the low-risk HPVs oncogenicity.

**Aims.** To update the pathogenic mechanisms driving the macroscopic and histological features of oral papillomatosis having HPV infection as the main etiological factor. High-risk HPVs are those leading to oral cancer, so we paid particular attention to defining them regarding their natural infection (transmission, clearance or persistence) and ability to interfere with the local immune pathways. We focused also on HPV autophagy and oncogenicity. The clinical aspects of benign and intermediate papillomatous lesions and the laboratory tools useful to the physician for an etiologic accurate diagnostic were also addressed.

## 2. Research Methods

For this review, we conducted an electronic literature search in the PubMed, Science Direct, Google Scholar, Scopus and Web of Science databases related to human oral papillomatosis and HPV oral papillomatosis, using as keywords: oral papilloma, HPV infection, HPV innate local immunity, HPV oncogenicity, Toll-like receptors, HPV vaccination, and HPV oral clinical manifestations in all the relevant combinations. Case reports, case series, and literature review-type articles from the earliest data available to February 2022 were included in our research.

## 3. HPV Morphology

Morphologically, these non-enveloped viruses have a hemispherical shape and dimensions between 50 and 60 nm [2,29,30]. Even in a small virus of 8 Kpb, the HPVs are composed of eight genes, called open reading frames (ORF), six early (E1, E2, E4, E5, E6 and E7) and two late (L1 and L2), necessary for different stages of the virus cycle, and a non-coding long control region (LCR). The LCR contains a DNA replication origin and binding sites for host cell transcription factors and for the viral E1 and E2 proteins that control viral replication and gene expression. E5, E6, and E7 encode three viral oncoproteins, E6 and E7 being the dominant oncoproteins of high-risk HPVs that modulate the transformation process [1,2,29]. The morphology of a high-risk HPV is displayed in Figure 1.

The six early genes (E1, E2, E4, E5, E6 and E7) are depicted in blue and the late L1 and L2 in yellow. The free space between L1 and E6 represents the long control region (LCR).

The main role of these proteins during the HPV cycle is to create a permissive cellular milieu for viral replication: the induction of DNA replication machinery, immune evasion and the downregulation of apoptosis. The viral oncogenes achieve these conditions by distorting the p53 and pRb cellular regulatory pathways [37]. E1 and E2 encode two regulatory proteins modulating the transcription and viral replication and two structural proteins; the late genes L1 and L2 encode the viral capsid proteins necessary for the final viral assembly and mediate the viral entry into the next infected host cell [2,14,16,38].

Figure 2 summarizes the roles of high-risk HPV main gene products.

## 4. HPV Transmission

HPV infection of the oral mucosa is determined by sexual or non-sexual transmission. The oro-sexual practices are incriminated in the sexual transmission, correlated with sexual behavior (number of partners, age of sexual live onset [39,40]). Interestingly, a horizontal transmission via saliva from mouth to mouth is also possible, as oral sex is not compulsory for HPV infection from a positive partner [1]. It seems that the bifocal transmission of oral and vaginal mucosa between partners is not usual, because they have different susceptibility, and the protective attribute of the saliva may be considered [39]. Vertical transmission is also accepted, the newborns with HPV-positive mothers having 33% more risk for HPV infection [1].

Oral papillomatous lesions can be transmitted from the skin to the oral mucosa. An eloquent example of this situation is verruca vulgaris, frequently found in children [40,41].

## 5. HPV Natural Infection

Both low-risk and high-risk HPVs have an affinity for the squamous epithelium of the oral and cervical mucosa and infect epithelial cells (Kcs). HPV’s capacity to immortalize in vitro in oral Kcs allows the extrapolation of the biological infection from the genital mucosa to the oral one [39].

The mucosa lining the oral cavity is formed of an epithelium sustained by a basal membrane (BM) which joins the epithelium with the underlined chorion or *lamina propria*. An orthokeratinized epithelium, similar to the skin, is present in the gum—the gingival mucosa—and at the level of the hard palate [42,43,44]. In the basal and intermediate layers of the epithelium, cells involved in immunity may be present through the Kcs, but these are numerous in the chorion. The oral epithelium from the jugal and labial mucosa and the soft palate is non-keratinized [42,45,46].

HPVs are exclusively intraepithelial pathogens, and an epithelial wound creating a discontinuity through the Kcs is necessary in order to allow the access of virions to the basal membrane and the profound cell layer [29,47]. Since the basal Kcs of the stratified squamous epithelium are the only ones able to divide, they are the initial target of HPV infection. For low-risk HPVs, which do not stimulate keratinocyte proliferation, this is a good hypothesis, but for high-risk HPVs, which interfere with cell proliferation, it is less clear. For a better understanding of high- and low-risk HPV infection and evolution, see [2].

A study on a mouse model highlighted, surprisingly, that HPVs do not initially bind the Kcs in vivo, but first the heparan sulfate proteoglycans (HSPGs) of BM exposed after the upper epithelial trauma via the L1 major capsid protein. HSPG is considered the first receptor. It also emphasizes the natural dependence of some viruses, including HPVs, on HSPGs used as hijacked receptors in order to bind to host cells. This typical interaction occurs through electrostatic interactions between the cell membrane and the virus. After binding to HSPGs, the mature virus particle undergoes a series of conformational changes to enter the host cell [48,49]. Following these conformational changes, viral capsids can be transferred to a non-HSPG epithelial cell receptor. To date, after binding HSPGs, HPV-16 interacts with the EGFRs present on the plasma membrane and produces an important reconfiguration of the host cells [49]. Once internalized by a micropinocytosis-like mechanism involving actin, the virus traffics through the endolysosomal system and reaches the nucleus [1].

Different viral genes are expressed during the cell-hosting virus superficialization [29], and the expression of viral proteins seems to be regulated by the differentiation of Kcs. Virus multiplication takes place in the internal layers of the epithelium driven by E6-E7 genes, key regulators of cell cycle progression, but genome amplification with viral protein production is noted in the upper epithelial layers, *stratum spinosum* and *stratum granulosum*, under the E4 gene control [2,50]. Here, the number of DNA copies is highly increased, up to a thousand/cell [47], and finally, new virions are packed and leave the epithelium under the control of early E4 and late L1 and L2 genes [1,29].

The accomplishment of a virus cycle takes time even in optimal conditions, and depending on the oral topographic area, the delay from the infection to the virion release is about 20 days, the time for keratinocyte daughters to reach the *stratum corneous* from the basal membrane [50].

The pathogenesis of low- and high-risk HPVs is notably different, even in the presence of the same gene products: low-risk HPVs produce a rapid progeny and a large lesion of the mucosa and induce the local immune response and inflammation; E6 and E7 proteins display low activity of transforming the host genome and lack the capacity to produce genomic instability [47]. Depending on the high-risk or low-risk virus genotype, the host immune response, the epigenetic conditions and the topographic region of the oral epithelium, the virus could get cleared, could remain latent or self-servingly drive the cell cycle to replicate and onset the pathological changes.

## 6. HPV Clearance

Clearance is a feature of the majority of HPVs. Usually, 63% to 70% of viral infections cleared rapidly within 12 months in response to an efficient immune system [51]. For most subjects, HPV-associated lesions are cleared within 1–2 years [52]. Virus clearing is associated with sexual behavior, while no epidemiologic link between the oncogenic or non-oncogenic types of viruses and age was noted. The risk of HPV infection decreases with age and is probably linked to a higher presence of HPV antibodies [53]. The median time for clearance of any type of HPV infection was significantly longer in young men as compared to other age groups [40]. Other studies state that the median time of HPV-16 clearance is nearly two times longer than with other oncogenic HPV types [54].

The resolution of infection requires the cross-priming of dendritic cells (DCs) followed by T-cell infiltration into the site of infection and the shut-off of viral gene expression [29]. An interesting study of the imbalance between the effectiveness of the immune system versus the stochastic dynamics of basal stem cells suggests that the chance of the division of basal Kcs plays a critical role in the elimination of HPV-infected cell clones. In subjects with a normal immune capacity, the immune response may contribute to less than 20% of the viral clearing, the rest being ensured by the random succession of symmetric and asymmetric stem cell mitosis [55].

Virus clearance could be illusory as lesion regression is not associated with massive apoptosis or cell death. The lesion is cleared by the replacement of actively infected cells with ‘apparently normal’ cells as the basal cells continue to divide. These ’apparently normal’ cells may still contain viral genomes but without concomitant viral gene expression, and, in animal models, it has been suggested that the virus life cycle may become ‘reactivated’ depending on individual conditions [29].

## 7. HPV Persistence

It is a general agreement that the Kcs of the oral mucosa are the reservoir of HPV, and the most common behavior of the oncogenic HPVs, especially HPV-16 and 18, but also of the low-risk HPV-6 and 11, is the virus persistence [56]. In order to maintain itself inside the host cells, either viral DNA is integrated into the host genome as for high-risk HPVs, or it is maintained as episomes that tether to host DNA as for low-risk HPVs. The integration of HPV DNA alters the expression of key genes in the host and promotes genomic instability and oncogenic progression [56].

HPV-6 and 11 are responsible for RRP that rarely can convert to cancer [36]. It has been reported that this rare oncogenesis of RRP may be caused by the integration of the HPV gene into the host cell genome, thereby knocking out the host gene [57]. Research data provide evidence that HPV-11 poses a higher potential for malignant transformation [57]. The oncogenicity differences between these low-risk HPVs could be related to the ability of E6 and E7 proteins of HPV-11 to have a more efficient interaction with host cell “oncogenic targets”, than those from HPV-6 [36,58]. It demonstrated the involvement of E2 protein from high-risk HPV-16/18 and HPV-11 to maintain HPV DNA as “mini-chromosomes” in dividing cells through its direct interaction with the mitotic spindles [56]. Still, HPV-6 and 11 proteins have a different ability to interfere with tumor suppressors p53 and pRb and other cell cycle regulators than those from high-risk HPVs [36].

Among the risk factors for virus persistence, there are the immunosuppression of HIV-infected patients and renal transplant recipients and smoking [1], albeit recent studies have described smoking and HPV infection as independent risk factors for oropharyngeal cancer. Subjects with some gene polymorphisms of the major histocompatibility complex (MHC) are also more susceptible to HPV persistence and cancerization, as reviewed in [56].

Research data is not congruent that co-infection with various HPV types is a potential predictor of subsequent persistence of the HPVs [59]. Chronologically, virus persistence is related to the storage of the viral genome in the epithelial stem cells followed by virus reactivation at the site of the previous infection or in other regions [60]. In the nuclei of the Kcs from the epithelial basal layer, the viral genome is maintained as an episome or plasmid (a low copy number), meaning that viral gene expression is minimal [61].

The state of viral genome integration in the host cells, with no viral DNA replication but production of the oncoviral proteins and nuclear mutagenesis, is called “pseudolatency”, three oncogenic genes being expressed in this stage, E5, E6 and E7. E5 gene product stimulates EGF-mediated cell proliferation and affects apoptosis [62,63]. The E5 protein acts as a viroporin, affecting vesicle trafficking and cellular homeostasis, altering antigen presentation and inflammation [64]. This latent phase can persist for years [1]. In this phase, unlike the low-risk HPVs DNA, the oncogenic high-risk HPVs DNA is integrated into the Kc genome under the action of E6 and E7 that escaped the E2 protein control [47]. This is considered a canonic viral gene imbalance explaining the persistence and carcinogenesis of HPV [65]. The E6 and E7 proteins from low-risk HPVs shared many of the cellular effects of their counterparts from high-risk HPVs but to a generally lesser extent. An explanation for their different oncogenic potential could be related to the different efficiency of action on cellular targets. The research data showed that low-risk HPVs producing RRP might undergo mutations which lead to increased levels of E6 and E7. This could explain why, once penetrated into the lung tissue, these pathogens determine lung cancer [58].

Some events (epigenetic factors such as local irritation) may determine the virus from the proliferative zone of the epithelium to leave the stable replication to their entry into the vegetative mechanism [1], and, as the virus multiplies, to migrate with the Kcs through the epithelium surface [29,47]. This explains the fact that the oral mucosa, together with the ductal epithelium of the salivary gland, the epithelium of the tonsillar crypts and the epithelium of the gingival pockets may represent an infection reservoir for the remote carcinogenesis of tonsils and pharynx [1,39].

The production of the HPV-16 early E5 gene ensures the persistence of infection in immortalized HPV-infected Kcs (Kcs) by downregulating the expression of MHC-I molecules through retention in the Golgi apparatus membranes and the inhibition of their transport to the cell surface [66], reducing the susceptibility of Kcs to be cleared by the cytotoxic T lymphocytes (CTL) and Natural Killer (NK) cells [67].

It should be noted that, in contrast to cervix carcinoma and epidermis, in the oral mucosa, virus multiplication is achieved without the damage of oral Kcs, and consequently premalignancies associated with OSCC are unknown [65]. However, a reduced functional impairment of Kcs in advanced stages of oral papillomatosis may occur [68,69].

It is not evident that a particular area of the epithelium is more susceptible to becoming a transitional host or a long-term reservoir for HPVs. A surprising finding is that among the HNSCN etiology, HPV’s prevalence in the oropharyngeal location is significantly higher in palatines tonsils (61, 8%) and the base of the tongue/lingual tonsil (49, 45%) [1], both areas included in the Waldayer‘s lymphatic ring. It seems that in this location, HPV infection may occur even in the absence of an anterior epithelial wound due to the particular structure of the epithelium, called improperly reticulated epithelium, as its cells and the underline basal membrane are discontinuous. It was speculated that the evolution of an HPV infection into a squamous cancer may be caused by the intimate relationship between epithelial and immune cells [70].

A second interesting site, with a particular histological organization and a huge involvement in the gingival homeostasis, able to host a latent HPV, is the epithelium forming the wall of the gingival pocket. As in the tonsillar *criptae*, the Kcs dynamically multiply with the faster turnover of the oral cavity, 4–6 days, loose packing and the numerous intercellular spaces created filling with crevicular fluid [71]. The access of HPV through the BM is facilitated and its integration into the genome may be more efficient. The coexistence of epithelial and immune cells is a feature of both locations. Underpinning this idea is the observation that oncogenic HPVs were found concealed in the junctional epithelium of the dental pocket in 26% of the analyzed cases of periodontitis. Whether HPV is a trigger of the periodontal disease or is installed in the slipstream of other pathogens enhancing the local inflammation is to be studied [72,73]. Epidemiologically, the gum is the most common site of OSSC HPV positive [8].

## 8. HPV and Local Immunity

The fate of the HPV virus infection of the mucosa is determined by its ability to hide itself and survive and the inability of the local cells to catch and neutralize the viral antigens. The balance between these conditions defines the virus clearance, persistence or cancerization. Some biological mechanisms used by HPV in order to escape the immunity cascade are summarized in [1,50,74].

The effective evasion of all levels of innate immune recognition seems to be the hallmark of HPV infections. HPV globally down-regulates the innate immune signaling pathways in the infected Kcs; pro-inflammatory cytokines, particularly the type I interferons, are not released, and the signals for Langerhans cell activation and migration and consecutively for the recruitment of stromal DCs are diminished or abolished [29].

We intend a brief description of HPV influence on the main actors of the innate immune response in order to understand the heterogeneity of clinical aspects in HPV infection.

**Keratinocytes** (Kcs) targeted by HPVs are considered as nonprofessional antigen-presenting cells (APCs), defined as “immune sentinels”, because of the various cytokines produced in response to innate immune sensor signaling [75]. The local immune response is initiated by the Kcs and the professional APCs: (i) Langerhans cells intermingled with the Kc in the basal and parabasal layers of the oral epithelium and (ii) macrophages. Then, specialized immune cells, such as the lymphocytes (Lys) harbored in the epithelium and in the lamina propria, tailor and refine the immune response. Kcs activation is crucial for the stimulation of residential macrophages, DCs and T effectors cells. All these effectors are endowed with pathogen recognition receptors (PRR): (i) Toll-Like receptors (TLRs) that possess the ability to recognize pathogenic structures from viruses, bacteria and parasites defined as pathogen-associated molecular patterns (PAMPs) and initiate the cascade of specific immune response; (ii) Nod-like receptors (NLRs) recognizing not only PAMPs but also damage-associated molecular pattern molecules (DAMPs) related to intrinsic danger such as unscheduled cell death, necrotic cells and tissue stress, reviewed in [76,77,78,79]. Potentially, HPV PAMPs are the L1 and L2 capsid proteins and the double DNA genome, reviewed in [80].

TLRs are particularly important in establishing the evolution of lesions because they have the role to guide the individual’s acquired immunity in order to neutralize viral HPV infection. The activation of TLRs on Kcs leads to the production of type I (α and β) interferons (IFNs) and enhances the cytotoxic response of Th1 lymphocytes [50]. A reduced expression of TLRs indicates the progression of the disease, while their increased expression is suggestive of the regression of the pathology [81]. Except for TLR7 and TLR8, the others are found in Kcs. TLR9 hosted in the cytosolic endosomal compartment of Kcs, neutrophils, plasmacytoid dendritic cells (pDCs), B and Th1/Th2 lymphocytes, fibroblasts and endothelial cells is able to recognize PAMPs that have already passed the cell membrane, such as viral double-stranded DNA [78]. HPV can avoid immune recognition by downregulating TLR9 through E6 and E7 proteins; in low-risk HPV genital infection, TLR9 repression does not occur. TLR7 and TLR9 signaling can induce not only inflammatory cytokines secretion but also type I IFNs, including IFN-α and IFN-β [79]. The effectiveness of TLRs activation during HPV evolution was studied by [82] who demonstrated that RNA expression of 1–3 and 6–9 TLRs was increased in cytological samples from patients with healed lesions; Britto et al. found their decreased levels in patients with progressive infection [83]. Their study conducted on cervical tissue and blood samples from HPV+ and HPV− patients evaluated the expression levels of genes involved in innate immune responses and cell adhesion. HPV-infected patients expressed higher levels of TLR9 and lower levels of PRR that recognize RNA (TLR3, TLR7 and MDA5/IFIH1). The authors concluded that HPV infection leads to changes in the innate antiviral immune responses: it increases the levels of DNA recognition receptors, and it decreases the expression of TLRs that recognize RNAs [83]. HPV+ samples displayed also a reduced expression of genes for *adherens* and tight junctions favoring other viral infections, i.e., HIV infection [84].

The natural life cycle of the HPV virus is entirely intraepithelial, and the infection does not produce local inflammation, cellular apoptosis or cytolysis, viremia after virus release with a successful getaway of innate immune recognition and consecutively the activation of adaptative immune reaction. Viral regression is associated with the successful priming of an adaptative immune response after antigen takeover by the APCs [80].

In HVP infection, Kcs response as APCs are abolished, and they are not able to activate the other APCs or synthesize regulatory and signaling molecules, such as IFNs, TNF-α and intercellular adhesion molecules as a response to the activation of TLRs [29,50]. IL-1 and type I IFN are the keystones of the keratinocyte’s immune response for the activation of Langerhans cells, macrophages, T helper lymphocytes and, respectively, cytotoxic TH1 lymphocytes [50,85,86]. IL-1 release occurs only after the inflammasome activation [50,87]. Under resting conditions, Kcs secrete IFN-k, a new type of I IFN which is constitutively expressed, unlike IFN-α/β, which is secreted only after stimulation. IFN-k has been shown to signal through the type I IFN receptor and to promote the upregulation of IFN-stimulated genes (ISGs) that can then promote antiviral immunity [88]. To evade IFN responses, HPV oncoproteins E6 and E7 repress both the basal and agonist-induced transcription of IFNs and ISGs. IFN-k in particular seems to be a target of HPV epigenetic repression both in vivo and in cell cultures [19,89,90,91,92]. The production of IFN-α and IFN-β, which are known to be endowed with antiviral immune responses and antiproliferative and immunostimulatory properties, is induced after the TLR3 and TLR9 activation of Kcs.

In the effective innate immune response, consecutively to NLRs stimulation, Kcs secrete a lot of cytokines IL-6, 10, 18, TNF-α and IL-1, the vault stone of the downstream immune activation of DCs and T helper lymphocytes. In HPV episomic infection, the reduced number of viral genomes and lack of cellular necrosis decrease NLR proinflammatory signals [50]. The central role of Kcs and their relation to other immune cells are depicted in Figure 3.

**Professional antigen-presenting cells.** Dcs and macrophages act as professional APCs as they activate the T lymphocytes, the effector cells of the acquired immune system.

The 1–8 arrows: Biological communication ways affected by the presence of the HPV in Kcs. (1). The innate immune response is principally affected by the choking of TLRs expression, with the diminished synthesis of interferons and ILs acting on effector cells. The absence of TNFα inhibits the synthesis of MCP1 by Kc, and the chemotaxis of MΦ is impaired (3). MΦ may be also directly infected by the Kc without the activation of immune phagocytosis (4). (2) and (7). The action of NK and NKT cells is diminished by the lack of necrosis and inflammation and the loss of Cd1 receptors. The functions of dendritic cells are affected by Kc in many ways, including (5) the failure to synthetize CCL20, so their chemotaxis in the epithelium and connective tissue is impaired, and (8) the suppression of Ecad and the shortening of the dendritic cells’ presence in the infected tissues. Infected cells escape the recognition by Ly specialized cells due to the cessation of HLA synthesis and expression and the impairment of immune cell cooperation. The viral genes involved are highlighted.

**Macrophages** hosted in the lamina propria of the mucosa play a pivotal role in innate immunity. Tissue-resident-macrophages are activated by TLRs binding to viral components and as a response, they release several types of inflammatory cytokines, chemokines, and IFNs. One of the chemokines active on macrophages is monocyte chemotactic protein-1 (MCP-1), released by Kcs in the presence of TNF-α, but such secretion is downregulated by the HPV16 E6 protein. Activated macrophages kill HPV-infected cells via TNF-α secretion or antibody-dependent cytotoxicity [93].

The main effect of HPV E6 and E7 proteins on macrophage action against HPV infection is the inhibition of macrophage chemotaxis to the site of viral infiltration [94,95].

Another interesting pathogenic way of HPV infection is that the viral particles end up infecting macrophages. In a very elegant experiment, Cao et al. demonstrate that macrophages take up microparticles containing HPV leading to their infection. Such microparticles with a 0.1–1 µm diameter are released by the first infected Kcs and enter the macrophages through a non-phagocytic pathway, triggering the transfer of HPV into the macrophage cytoplasm by avoiding the phagolysosome synthesis and the virus lysis [96]. This interesting research may provide new insight into the persistence of the HPV infection.

However, even if they are mobile and major scavengers responsible for the phagocytosis of apoptotic cells and maintenance of local homeostasis, macrophages do not specifically migrate to the draining lymph nodes (LNs) and are less efficient in antigens presentation to T cells than the DCs [94].

**Dendritic cells** are generally divided into two classes: conventional DCs (cDCs) and plasmacytoid DCs (pDCs) [97]. In the oral epithelium, the cDCs harvested between the Kcs of the basal and profound parabasal layer are called Langerhans cells (LCs) by analogy with those from the epidermis, and those located below the basal membrane are simply called dendritic cells (DCs), considered the equivalent of dermal DCs. LCs and DCs differ by some markers: langerin is considered a specific marker for LCs from the mucosal epithelium [98], while CD206, CD209 and the mannose receptor are markers for DCs resident in the lamina propria [99]. pDCs are difficult to detect in peripheral tissues; they circulate in the blood and can be found in peripheral lymphoid organs [100].

Through the *lamina propria*, DCs migrate to the local lymph node. Antigen presentation in lymph nodes is necessary for the initiation of the acquired immune response by activating T helper (Th) CD4+ T cells differentiated into various subtypes: Th1, Th2, Th17, and T regulatory cells (Treg) [77,80,101].

In their immature state, DCs are the only mobile professional APCs that efficiently cross-present cell-associated antigens which refer to their ability to transfer exogenously derived antigens directly to the MHC class I and transport processed antigens to the neighboring lymph node in order to present them to CD8+ T cells [102].

LCs’ migration depends on the environmental chemokines and their specific receptors. The pair C-C motif chemokine ligand 20 (CCL20)-chemokine receptor 6 (CCR6) is responsible for the chemotaxis of DCs and plays an important role in the mucosal surfaces under homeostatic and inflammatory conditions [103]. CCL20 is a chemokine that is formed and secreted by Kcs, and LCs can express the CCL20 receptor, CCR6. The expression of CCL20 results in the migration of immature LCs to sites of inflammation in the epidermis [104]. The downregulation of CCL20 leading to LCs decline is a key factor in HPV lesions. High-risk HPV-type lesions might inhibit the chemokine CCL20 through E6 and E7 to escape the immune response. In vitro studies demonstrate that the inhibition of CCL20 expression is related to the downregulation of NF-kB signaling [105].

Several data suggest that the E-cadherin-mediated contact between KCs and LCs is important for the immune response during chronic HPV infection. The adhesive molecule E-cadherin, secreted by the KCs, mediates their adherence and allows LCs to remain longer in contact with the antigen [106]. In high-risk HPV infection, E7 protein seems to be responsible for the suppression of E-cadherin expression and LCs’ mobilization from the epithelium [107]. Significantly, the accumulation of LCs in cervical biopsies is associated with the clearance of cervical HPV infection [108].

CCR7 expressed by phenotypically mature DCs plays a role in mediating DC trafficking in response to the lymph node-homing CCL19 and CCL21 [109,110,111]. CCR7 expression is downregulated in HPV tumors, resulting in the impediment of their migration to the chemokines residing in the lymph nodes, an action necessary for initiating adaptive immune responses.

It is emphasized that mainly E5 and E7 of HPV16 and HPV18 are able to interfere with MHC class I (HLA—A, B, C) and also with MHC class II (HLA DR) and escape the recognition of T cells [112]. Effector T cell activation, either cytotoxic CD8+ (CTL) or helper CD4+ (Th), against an antigen presented by the APCs depends on the simultaneous presence of MHC or HLA (Human Leucocyte Antigen—in humans) restriction molecules, while NK cells are not restricted being able to kill in a non-specific manner, not requiring prior sensitization to antigen [113]. Not only the density but also the LCs’ maturation is compromised in HPV infection. As demonstrated, the expression of MHC class I and MHC class II and langerin are significantly reduced in HPV-positive cervical lesions compared to the normal epithelium. The downregulation of MHC class II molecules reduces the capacity of DCs to trigger the antigen-specific T cells. However, the overexpression and polymorphism of HLA G, a nonclassical HLA class I, with the capacity of suppressing the cytotoxic activity of CTL and NK cells was incriminated in HPV cervical cancer, but no correlation with HNSC was noticed. The increase in HLA type II is probably induced by IFN-γ augmentation mediated by Lys CD4+ and CD8+ [114]. In conclusion, LCs suffer a decrease in number, display reduced mobility and also a lack of maturity during HPV infection.

Additionally, the low-risk HPVs’ dysfunctional activation of acquired immunity by APCs was demonstrated to suppress HPV clearance and perpetuate the infection [36].

**NK cells and NKT cells.** NK cells, a population of Lys, are key effectors of the innate immune system, known to act as a bridge between the innate and adaptative immune response. When NK cells encounter their own healthy cells expressing normal MHC molecules, they are in a resting state; when modified cells with abnormal expression of MHC molecules are detected and surrounded by enough activating ligands, NKs immediately identify and eliminate virus-infected transformed cells activating mechanisms such as cytotoxic particles, cell apoptosis and antibody-dependent, cell-mediated cytotoxicity (ADCC) [115]. NK cells have a huge efficacy on some pathogens (such as herpesviruses—HHVs) which specifically down-regulates MHC class I molecules to escape cytotoxic T cells surveillance. Besides its inhibition of MHC class I expression, studies indicate that the E7 protein of HPV escapes the immune response of cytotoxic T cells by two mechanisms: (i) interaction with antigen processing-associated transport protein (TAP)-1 to prevent intracellular antigen peptide delivery following antigen presentation [116] or (ii) the imbalance of cytokines and chemokines secretion to activate other immune cells, such as DCs, T and B lymphocytes [117]. NK cells differentiation and activation could be suppressed by the downregulation of IFN-α [118] and upregulation of IL10 [119], as found in HPV mucosal lesions.

NKT cells are a subset of T lymphocytes that express the surface receptor characteristics of the T and NK cell lineages. They are subclassified in invariant and semi-invariant NKTs. NKT cells, similarly to conventional T lymphocytes, express a T-cell Receptor (TCR) [120], but unlike conventional T cells, which act on antigens in the presence of MHC class I or class II molecules, they react with lipid or glycolipid antigens presented by the MHC class I-related glycoprotein CD1d. A loss of CD1d downregulates the antiviral response of NKTs in the onset of the immune activation [47]. Following the recognition of CD1d, invariant NKTs rapidly secrete cytokines and activate the effectors of both innate and acquired immune responses, and simultaneously acquire cytotoxic activity. HPV E5 protein downregulates Cd1 receptor, evading HPV-infected cells from NKTs’ action [47].

The imbalance in the immune response is relevant to both high- and low-risk HPVs as demonstrated by the progression of a respiratory tract infection with a low-risk HPV to RRP [58].

These main signaling immune pathways suppressed after the infection of Kcs by high-risk HPVs are summarized in Figure 3.

## 9. HPV and Autophagy

Autophagy is a self-consumption cellular mechanism, important for maintaining the internal homeostasis and the functionality of the cells, which may be activated in physiological or pathological conditions, initially described in cellular starvation and the recycling of macromolecules. Following infection, autophagy can help to eliminate invading intracellular microbes and damaged proteins and sustain the organelles’ function as a cellular defense mechanism [121].

Autophagy is under the control of more than 30 autophagy-related genes (*Atg* genes); it consists of the lysosomal digestion of the proper organelles, such as mitochondria (mitophagy), endoplasmic reticulum (ER-phagy) or pathogen-like viruses or bacteria (xenophagy) in a succession of activities: the phagophore initiation, elongation and fusion of the autophagosome with the lysosome which results in an autolysosome. Autophagy primarily acts as a protective mechanism that may prevent cell death and contributes to the regulation and function of innate and adaptive immune responses. During infection, it plays an important role in the innate response to degrading intracellular bacteria and viruses: in adaptive immune responses, autophagy is involved in antigen presentation and lymphocyte development [122]. Autophagosomes can fuse with HLA class II loading compartments.

Each phase of autophagy is ordered by specific genes forming protein complexes whose activity is regulated by the nutrient condition through the mammalian target of rapamycin-mTOR, the upstream cellular sensor [24]. The nutrient condition and growth factors in both the external and the internal cellular milieu promote mTOR activation with cell growth and the blocking of autophagy. The inhibition of mTOR leads to the initiation of the phagophore by the activation of kinase ULK1 [62,123]. The phagophore membrane expansion is commanded by several genes, of which beclin-1 [49] is the most studied. The formed autophagosome moves through the microtubules up to the perinuclear region and fuses with the lysosomes. The fusion step is very precisely coordinated by three sets of protein families in cooperation with the beclin-1 complex [124]. The substances resulting after the lysosomal digestion are used by the cell to its own benefit [125].

Recent studies have elucidated the mechanisms by which intracellular bacteria and viruses are targeted by autophagosomes for degradation, as well as the mechanisms by which successful intracellular pathogens evade or co-opt the autophagy pathway to their own advantage [122,126,127]. HPV binding on Kcs and internalization in the incipient phases of infection manipulate cell autophagy. Interaction with HSPG triggers some signaling pathways in the host cell: HPV-16 interacts with EGFR on the cell membrane, and finally the activation of the mTOR pathway suppresses autophagy [122,127]. The degradation of the HPV capsid inside the cell is delayed, and thus the virions are protected from rapid degradation, and the lifespan of HPV inside the cells is prolonged [24,49]. HPV-16 E5 protein reduces the formation of the autophagosome by functionally inhibiting the activity of the p53 gene which affects the transcription of some ATGs, i.e., beclin 1 [128]. In the following steps of the carcinogenetic process, the HPV-16 proteins E6 and E7 affect the fusion of the autophagosome with the lysosome [49]. Later, autophagy is restored in order to remove the stress tissue conditions induced by cancer (hypoxia, inflammation, nutrient deprivation) and to ensure cancer progression [129].

## 10. HPV Oncogenicity

Oral cancer is a serious and growing problem worldwide. Oral and pharyngeal cancer, grouped together, is the sixth most common cancer in the world. The annual estimated incidence is around 275,000 for oral and 130,300 for pharyngeal cancers excluding nasopharynx, two-thirds of these cases occurring in developing countries [130]. The potential for malignancy of oral papillomatous lesions is largely influenced by the presence of HPV viruses, especially HPV-16—found in almost 90% of the HPV+ oropharyngeal cancer [16]—and HPV-18, but also HPV-31, 33 and 52 are all sexually transmitted [18]. Infections by low-risk HPV types such as 6 and 11 are rarely found in HNSCC [58].

The development of HPV-induced cancer is related to the ability of the virus to evade immune detection for a prolonged period in order to favor the accumulation of genetic abnormalities in the nucleus of host cells. According to Syrjanen and colleagues [131], a patient diagnosed with oral papillomavirus (HPV) is four times more likely to turn a benign oral lesion into a malignant lesion.

The risk factors of HNSCC HPV negative are represented mainly by chronic alcoholism, cigarette smoking, inadequate nutrition, poor oral hygiene and also infection with other microorganisms such as Candida albicans and the Epstein–Barr virus [12], UV and ionizing radiation and also genetic predisposition [4,132,133,134]. Actually, a change in the balance between HPV-negative and HPV-positive HNSCC was noticed, in part because of the decreased use of tobacco and alcohol and increased awareness and enhanced molecular diagnostic methods of HPV infection [15,135].

Interestingly, only the squamous cell carcinoma of the oropharynx may be directly linked to the HPV infection, while HPV has a minor involvement in the mouth cancerous pathology [37]. The difference between the clinical and molecular phenotypes of HPV-positive or HPV-negative HNSCC emphasized the importance of understanding the potential of the carcinogenesis of HPVs, corroborating the biologic condition and genetic mutations of the infected subjects. Low- and high-risk HPVs infect and replicate in the same way when encountering similar structures of the mucosa and needing to convert the same cellular defense mechanisms. Therefore, it is surprising that their cellular targets and induced pathologies are considerably different [37].

The involvement of some pathogenic pathways is presently debated, one using the p16 protein accumulation as a main marker of canonical HPV infection and the other related to Retinoblastoma protein (pRb) tumor suppression. P16 protein is a product of p16INK4a, a tumor suppressor gene encoded by the CDKN2A locus, located on 9p21 [136,137]. P16 is a cell cycle protein that arrests the cell cycle progression and normally functions as a checkpoint at the G1/S cell cycle stage, expressed at a very low level in normal cells. The increased expression of p16 occurs as a result of the functional inactivation of the pRB commanded by HPV E7 protein. In HPV-unrelated cancer, there is no perturbation of the pRb pathway, and the product expression of the p16CDKN2A is low. The upregulation of genes encoded for p16 requires the transcription of E7 oncogene, therefore the squamous carcinoma where p16 protein overexpression is noted is supposed to result in an action of HPV; as a consequence, p16 is considered a biomarker for HPV oncogenic action and a surrogate marker for HPV infection [137]. Nevertheless, the presence of p16 protein is not mechanistically linked to all HPV-positive cancer. A cohort study on HPV-positive and HPV-negative HNSCCN located in various topographical oral regions demonstrated that p16 prevalence is different between the oropharyngeal location and the non-oropharyngeal location, being significantly associated with the oropharyngeal region, smoking and marital status. Authors demonstrate also that p16 status is an important prognostic marker for both the oropharyngeal and extra-oropharyngeal oral cell carcinoma, and a distinct subgroup concerning the presence of p16 must be considered—a p16-positive/HPV-negative group, in which all HPV genotype markers are absent [137].

The HPV oncoproteins E6 and E7 target the same pathogenic pathway with different mechanisms, showing a strong corroborating action in promoting cellular transformation in the field of the cancerization and neutralization of the immune response [138,139]. Thus, HPV E6 and E7 are the keystones of the carcinogenetic outcome of HPV infection, as they abrogate the tumor suppression functions of p53 and pRb [137,140]. E6 protein binds to p53, a protein product of the TP53 gene, a transcription factor inducing cell cycle arrest and apoptosis in response to DNA damage, “guardian of the genome” and one of the most mutated tumor suppressor genes in cancer [16,141,142]. The mutation of p53 impaired the control of DNA replication with uncontrolled cell replication and a loss of cell apoptosis [143]. In HPV-positive HNSCC, the TP 53 gene is rarely altered [12], but HPV-16 and 18 facilitate the rapid degradation proteasome-dependency of the tumor-suppressor protein p53 via the formation of a trimeric complex including HPV oncogene E6, p53 and the cellular ubiquitin ligase E6AP. The complex formed induces the ubiquitination of p53 and stimulates its degradation [144,145]. The literature data showed that E6 and E7 proteins from low-risk HPVs could also interact with pRb and p53 but less efficiently, proving that they have different mechanisms to abrogate their functions [58,146].

The high-risk HPV E6 protein promotes cell proliferation and activates telomerase, and HPV E7 induces the entry into the S phase of the cell cycle [135,136].

HNSCC associated with tobacco and alcohol abuse has a worse prognosis due to the perturbation of the cell cycle regulation, the loss of p16 expression by deletion, mutation or hypermethylation resulting in p53 overexpression and causing the limitless replicative potential of head and neck cancer [33,147,148]. In a cohort study, Chung et al. emphasized that the role of p16 protein as a potential prognostic biomarker in non-HPV OPSCC tumors has to be considered only if each anatomical location is separately evaluated because important biologic and anatomic heterogeneity is present [149].

In direct relation to the action of p16 and the abrogation of the p53 tumor suppressor, the HPV oncoprotein E6 contributes to cell immortalization by telomerase activation. Telomerase is a ribonucleoprotein complex containing an RNA component (TR) and a catalytic protein (TERT) able to maintain telomere elongation, one of the mechanisms of telomere preservation in dysfunctional telomeres. Telomere dysfunction leading to genomic instability is considered an early carcinogenetic condition [144,150]. Short telomeres are often found in HNSCC precursor lesions and neighboring mucosa and trigger persistent DNA damage response (DDR). In addition to promoting cell cycle arrest, dysfunctional telomeres can also activate p53-dependent apoptosis, concluding in cell cycle arrest and apoptosis [139]. If p53 is absent, cell proliferation is allowed and the microenvironmental milieu sustains the survival of damaged cells and carcinogenesis [151]. Virally transformed human cells abrogate the p53 and/or pRB gene function and adopt a telomere-maintenance program: 80–90% of human tumors possess some telomerase activity and replicate indefinitely [144,152], while normal somatic cells do not express telomerase, so telomerase activity may be used as a tumor diagnostic marker [153].

Intending to improve the molecular diagnostic of HCSCC and the specific therapy, a distinct phenotypical population of cells was described—cancer stem cells (CSCs). From the heterogeneous population of cells forming epithelial cancer, it is the first implicated in tumor cell survival and progression, being highly tumorigenic as compared to the other cancer cells and is considered to be largely responsible for the biological characteristics of cancer—rapid growth, invasion and metastasis [154]. It was suggested that only the CSCs within the tumor can self-renew and proliferate extensively to form new tumors [155]. The identification of this tumor subpopulation would have important therapeutic and prognostic implications, but a single common marker failed to be found. Several molecular markers were described, especially in the diagnostic of oral squamous cell carcinoma: CD44, ALDH1, CD133, Oct3/4, Nanog, and Sox2 [156]. CD44 and CD133 were highlighted as markers for CSCs in HNSCC or specifically in OSCC, with different degrees of specificity [157]. Authors consider CD44+ more than CD133+ as a reliable marker for metastatic cells in HCSCC [158].

## 11. Clinical Aspects of Oral Papillomatous Lesions

If a clinical examination of a patient’s oral cavity reveals the presence of a tumor, this could be a papilloma of non-viral etiology or HPV-induced [4]. The histopathological examination performed after the surgical excision of the tumor confirmed the diagnotic of the papillomatous lesion, but the exact viral or non-viral etiology could be established only after performing a molecular test in order to detect virus presence because the macroscopic appearance and histological features of both types are similar (Figure 4). The main histological changes of the mucosa related to the epithelium regard hyperkeratinization, acanthosis and the presence of koilocyte cells; a normal relationship between the epithelium and lamina propria without infiltrative phenomena, exophytic or endophytic proliferation was also described [4,35,39,159].

According to the degree of malignity, three types of papillomatous lesions can be identified in the oral cavity: benign lesions, potentially malign disorders and malignant lesions. In this paper, we discuss the most important features regarding the first two.

**Benign epithelial lesions** in the oral cavity include squamous cell papilloma, condyloma acuminatum, skin verruca vulgaris, multifocal epithelial hyperplasia and papillary hyperplasia [160]. All these benign epithelial lesions have been attributed to HPV.

The most common benign lesion, reviewed in [132], is the **squamous cell papilloma**, with an incidence of 1:250 people, frequently affecting men aged between 30 and 50. However, the condition can occur at any age, even in children [1,161,162]. The HPV subtypes most often found in oral squamous cell papilloma are HPV-6 and HPV-11, topographically located on the labial mucosa, the palate and tongue.

Generically, at the oral level, ***verruca vulgaris*** appears as a solitary lesion, being characterized by “keratinized turns” and a very well-developed granular layer [163]. It is induced by HPV-2 and HPV-4, with a main prevalence in the vermillion border, labial mucosa and the anterior face of the tongue [132]. *Verruca vulgaris* is a sessile formation, while squamous papilloma has a pedunculated appearance. Microscopically, squamous papilloma has a poor representation of the granular cell layer compared to verruca vulgaris, and a finger-like aspect. Both have koilocytes in the prickle layer and no atypia accompanying the mitotic aspects in basal and parabasal Kcs, although authors claimed in some cases a reduced number or the absence of koilocytes [164].

***Condyloma acuminatum*,** most usually found in the genital rather than in the oral mucosa, is determined by HPV-6, 11, 16 and 18 subtypes and is located on the soft palate, lingual frenum and labial mucosa [132]. It occurs most frequently in young and adult patients because it is found simultaneously in the genital and oral mucosa and is considered a representative form of sexually transmitted HPV [1,165,166]. Nevertheless, additional routes of infection, such as through fomites, are possible [132]. Clinical studies have shown that this type of papillomatous formation is a multiple lesion; frequently, during the clinical examination of the oral cavities at least five formations of this type are identified. The clinical appearance of *condyloma acuminatum* varies: the shape can be sessile or pediculate, the color pink or white and the size proportional to the age of the lesion, but always significant compared to squamous papilloma. Koilocytes, a pathognomonic element for viral infection with HPV, are present, but acanthosis, important parakeratosis and bulbous rete pegs may also be highlighted by the microscopical examination [132,166,167]. Oral condyloma cannot be reliably distinguished from oral papilloma either histologically, clinically or by HPV genotyping, because HPV6/11s are the most prevalent types in both lesions [160].

**Multifocal epithelial hyperplasia** or **Heck’s disease,** reviewed in [168], is induced by HPV-13 and 32 subtypes and is located on the lining mucosa and on the tongue [132]. In the past, multifocal epithelial hyperplasia was identified and still remains predominant in Latin American indigenous children. Nowadays, it has been reported all around the globe, mainly affecting women and elderly subjects and patients diagnosed with HIV [169]. Recent studies have drawn attention to the fact that multifocal epithelial hyperplasia can be genetically transmitted as it was reported in isolated cases produced by the leukocyte antigenic subtype HLA-DR4 B1 *0404 allele [1,168,170]. The characteristic feature is the presence of mitosoid bodies in the basal layer—a pattern of degenerating chromatin that may mimic mitotic figures which, corroborated by koilocytosis, demonstrates the viral cytopathic effects [1].

Examining atypical and malignant papillary lesions for low- and high-risk HPV, it was demonstrated that simple squamous papilloma are rarely HPV-positive. Low-risk HPV-6 and 11 infection lead to papillary proliferations with clinically and/or histologically atypical features but with a benign clinical evolution, meanwhile potentially malignant lesions are not associated with low- or high-risk HPV [35].

**Potential malignant disorders.** Oral potentially malignant disorders (OPMDs) is a term which refers to a potential precancer stage of oral squamous cell carcinoma. Papillomatous lesions considered with the potential of malignancy are oral leukoplakia, oral erythroplakia, proliferative verrucous leukoplakia and oral lichen planus [39,131].

**Leukoplakia** consists of the presence of white plaques in the oral mucosa. In a cohort study of 890 leukoplakic and keratotic lesions tested for HPV, 25.4% were HPV-positive [171]. Clinically, leukoplakia can be subdivided into a homogeneous type (flat, thin, uniform white in color) and a non-homogeneous type. The non-homogeneous type has been defined as a white-and-red lesion (“erythroleukoplakia”), that may be either irregularly flat (speckled) or nodular. Occasionally, koilocytic changes in dysplastic lesions (“koilocytic dysplasia”) can be observed, apparently related to the presence of intermediate and high-risk HPV.

**Erythroplakia**, defined as “a fiery red patch”, cannot be confused clinically or pathologically with any other definable lesion and occurs mainly in middle-aged and elderly subjects. Microscopically, erythroplakia commonly shows at least some degree of dysplasia and often even carcinoma in situ or invasive carcinoma, more than 54.5% being HPV-16 positive [131].

**Oral proliferative verrucous leukoplakia** (PVL) is very rare, has recently been defined as a form of oral leukoplakia [172] or an independent entity, displays multifocal distribution and has a strong predisposition for oncogenicity. Because of the lack of specific histological features and of the progressive proliferative trend, this under-diagnosticated pathology is critical because, irrespective of the presence of dysplasia, it may progress into carcinoma [173]. PVL is clinically indistinguishable from the clinical aspect of verrucous carcinoma and is more prevalent among elderly women [174].

**Oral lichen planus** (OLP) is a chronic oral inflammatory disease of multiple disputed etiology, frequently defined as an autoimmune disorder because of the profound imbalance of lymphocytes and APCs. OLP affects the oral mucosa, the tongue and, to a lesser extent, the gums and lips [175].

By molecular methods, some studies have detected the presence of HPV in OLP in a range of 0.22% to 5% worldwide, 23% being positive for HPV-6 and 11, followed by HPV-16 [131]. The typical lesions of OLP are painful and persistent erosions (erosive LP) or diffuse erythema and the peeling of the mucosa (desquamative LP). In addition, Wickham striae may be present in a lacy or fern-like pattern [176]. The disease is characterized by a massive infiltration of lymphocytes, mainly T cells, beneath the epithelium, and an abnormal epithelial keratinization cycle: the rate of differentiation of stratified squamous epithelium is increased with either epithelial thickening or atrophy with or without ulceration [177]. The presence of activated LCs/DCs in the inflammatory infiltrate was demonstrated. Recent studies identified the accumulation of mature LCs and DCs in the submucosa of OLP biopsies, and these cells were co-localized with lymphocytes. Such cellular organization was proposed in order to allow the ongoing presentation of self-antigens to T lymphocytes that facilitate tissue destruction. pDCs also found in OLP are able to secrete IFN-α and contain granzyme B, important to induce the maturation of DCs and to synergize with lymphocytes in tissue destruction [97].

Genetic studies noted various HLA-based susceptibilities of LP in different populations: HLA-B27, HLA-B51, HLA-Bw57 (oral LP in English patients), HLA DR1 (cutaneous/oral LP) and HLA-DR9 (oral LP in Japanese and Chinese patients). The trigger for the onset of the lesion is environmental factors such as systemic viral infection, hepatitis C and members of the human herpesvirus (HHV) family, specifically HHV-6 and HHV-7 [178]. Additional environmental factors have been implicated in the development of OLP which include changes in the oral microbiome (e.g., *Candida* sp., various other bacterial infections) and dental metals precipitating allergic contact reactions [179].

Some researchers supported the idea that the chronic inflammation of OLP could promote the malignant transformation [180], the most significant predictor of which was the recurrence of OLP after treatment [166].

## 12. Diagnostic of Oral Papillomatosis

The presence of HPV may be suspected only in productive lesions with objective changes in the oral mucosa. In these cases, the histopathological examination highlights the presence of koilocytes, a pathognomonic element of HPV infection [132,181]. They are present on the surface of the spinous layer and predominant in the genital and the oropharyngeal mucosa [1,182,183], with a vacuolar appearance, a clear halo and often a pyknotic and hyperchromatic nucleus [184,185]. Koilocytes are considered transformed Kcs as a result of DNA damage and an accumulation of mutations [49,186].

An inconsistent presence of koilocytes in oral lesions of condyloma means it is difficult to distinguish between oral papilloma and condyloma [35,187]. Atypical clinical lesions, including those with larger size, irregular outline, progressive growth, wide and elongated rete ridges, koilocytosis and dysplasia of the epithelium that covers the papillary projections raised the possibility that these were lesions of condyloma acuminatum [35].

The precise etiologic diagnostic may be direct by identifying the virus or indirect by measuring the antibody concentrations in the biological secretions. Unfortunately, standard diagnostic protocols are not in use in oral medicine as they are in gynecological services. In cervix carcinoma, the premalignant stages are detected by staining with acetic acid while the standardized diagnostic techniques of the premalignancies of OSCC associated with HPV are missing [65].

The presence of HPV in the lesion may be searched by a variety of methods, including immunohistochemical staining for HPV common antigen, in situ hybridization (ISH) with DNA probes, polymerase chain reaction (PCR), and in situ RT-PCR [35].

Immunohistochemical methods are useful because they identify capsid proteins expressed in the late stage of the HPV life cycle. The only immunohistochemical marker with recognized diagnostic significance is p16 protein (INK4), a surrogate marker, whose overexpression in dysplastic/neoplastic oral or cervical histological samples is strongly associated with high-risk HPV infection, although it is not possible to rule out the simultaneous presence of other genotypes [15,35,39,188]. Immunohistochemistry used for p16 proved that α-HPVs were present only in a minority of benign verrucous and papillary oral lesions [160]. The detection of elevated p16 in HNSCC is considered a marker for the presence of a biologically active HPV infection and also a condition for a good prognostic [15,189,190]. p16^INK4a^ IHC is highly sensitive but moderately specific to diagnose HPV-transformed OPSCC when used as a single test [191].

The molecular detection of HPV, performed in tissue samples or in exfoliated cells, is increasingly used in clinical gynecologic practice for various purposes: either to identify women with cervical cancer risk, or to follow up the subjects after local treatment, and presently to screen the subject prior to vaccination. The molecular techniques, PCR and ISH, performed in tissue biopsies or in secretion are different in sensitivity and specificity and may be classified into: (i) target amplification assays/PCR, a highly sensitive and specific method, able to detect 10–100 HPV viral genome in 100 ng of cellular DNA. The SPF10-LiPA system, to identify a 65 bp fragment of the HPV genome, has been introduced, enabling successive PCR amplification and genotype identification [39,192]; (ii) direct hybridization assays—Southern blot hybridization and in situ hybridization are less sensitive and need a large amount of purified DNA; and (iii) signal-amplified hybridization assays, reviewed in [189].

The accuracy of the diagnostic of HPV causality seems to influence the therapeutic decision in OSCC, therefore some authors proposed algorithms for diagnostic techniques.

The determination of high sensibility, considered to be a gold standard, is the ratio of HPV E6/E7 mRNA expression which indicates the degree of active viral oncogene transcription inside the tumor cells [149]. Although this is a very laborious and out of the ordinary technique, it is important, as HPV-driven carcinomas critically depend on the carcinogenic action of the HPV E6 and E7 oncogenes indicating the transforming relevance of HPV [193,194].

The determination of HPV-16 and HPV-18 E7 transcript levels in HNSCC may also be correlated with the severity of the lesion [15].

In HPV-associated carcinomas, p53 is missing and the tumor suppressor protein p16 is overexpressed due to viral oncoprotein activity. The combination of p16 with the HPV DNA test is the most practicable test algorithm. Combined p16^INK4a^ IHC and HPV DNA PCR testing significantly enhances specificity while maintaining high sensitivity [191], the most suitable being p16 IHC followed by GP5+/6+ PCR on the p16-positive cases, because GP5+/6+ PCR allows screening for HPV infection and viral load quantification simultaneously [195].

The examination of saliva or oral rinses for HPV’s presence is an inexpensive, easy and noninvasive technique, valuable for prophylaxis or therapy monitorization. Mouthwash and the examination freshly by SPF10-LiPA are recommended as a screening method to identify the virus species [39].

Serological tests may be used as an indirect diagnostic method, but they are inaccurate in diagnosing a previous infection. The presence of antibodies is more valuable as presumptive diagnostic when risk factors exist or in the prognosis of the evolution. In most patients with HPV-associated OSCC, HPV-specific antibodies (E6 antibodies) are present in the plasma more than 10 years before the diagnosis of oropharyngeal cancers [196]. The detection of E6 and E7 antibodies correlates better with the patient outcome than the tissue HPV tests. However, some literature data contradict this hypothesis because anti-E6 and E7 antibodies are inconsistent, and often they cannot be assessed in order to diagnose and monitor the potentially malignant lesion [197,198,199,200,201]. Regarding the titer of anti-HPV antibodies, it was shown that it is significantly higher in serum compared to the secretions taken from the cervical level [201].

## 13. HPV Vaccines

HPV vaccination is nowadays recommended both for children and adolescents. The adults between 27 and 45 years who are not already vaccinated may decide also to get the HPV vaccine, but it is recommended to be tested for serum level of anti-HPV antibodies [201,202].

The ability of the major capsid antigen of HPV, L1, to self-assemble into highly immunogenic virus-like particles led to the development of three HPV vaccines (the bivalent, the quadrivalent and the nonavalent HPV vaccine) [201]. In contrast to natural HPV infection, vaccination induces high-quality and sustained serum antibody titers and confers protection against persistent incident infections and pre-malignant neoplasia. The assessment of the antibody titer showed that the natural immunity obtained after passing through the disease had a lower titer of anti-HPV antibodies (even 100 times lower) compared to the immunity acquired after the administration of the HPV vaccine [201,203].

The main HPV vaccines used worldwide are Gardasil 9 and Cervarix [204,205,206,207]. Gardasil 9 is a nonavalent vaccine to prevent infection with the following HPV virus subtypes: HPV-16 and HPV-18 (increased oncogenic risk), but also for HPV-6, 11, 31, 33, 45, 52 and 58 with low oncogenic risk [206]. Cervarix is a bivalent vaccine and is effective in protecting against infections with HPV virus types 16 and 18 [206,208].

Numerous studies that have been conducted over time have demonstrated the prophylactic nature of vaccines, but also their safety [209,210,211]. As with other vaccines, a number of side effects have been reported, most of which are characteristic of vaccines in use for other conditions [211].

## 14. Conclusions

An etiological factor of major importance for oral papillomatosis is HPV infection, but other non-viral factors that exert detrimental effects on the oral mucosa are involved. The evolution of natural HPV infection depends on the virus type, oncogenic or non-oncogenic, but also on the particularities of the innate immune response whose main effectors are subjected to systematic suppression. The fight between the virus and the aggressed epithelium determines the evolution of the infection: clearance or persistence and oncogenicity. Understanding how local immunity works guides the physician in establishing a correct diagnosis and in applying personalized treatment to the patient. A complete diagnosis of oral papilloma can be established through a good knowledge of etiological and epidemiological factors, clinical examination and laboratory tests. It is noted that OSCCs have acquired an increased incidence, and there is a need for the development of more accessible methods for an earlier precise diagnosis.

## Figures and Tables

**Figure 1 medicina-58-01103-f001:**
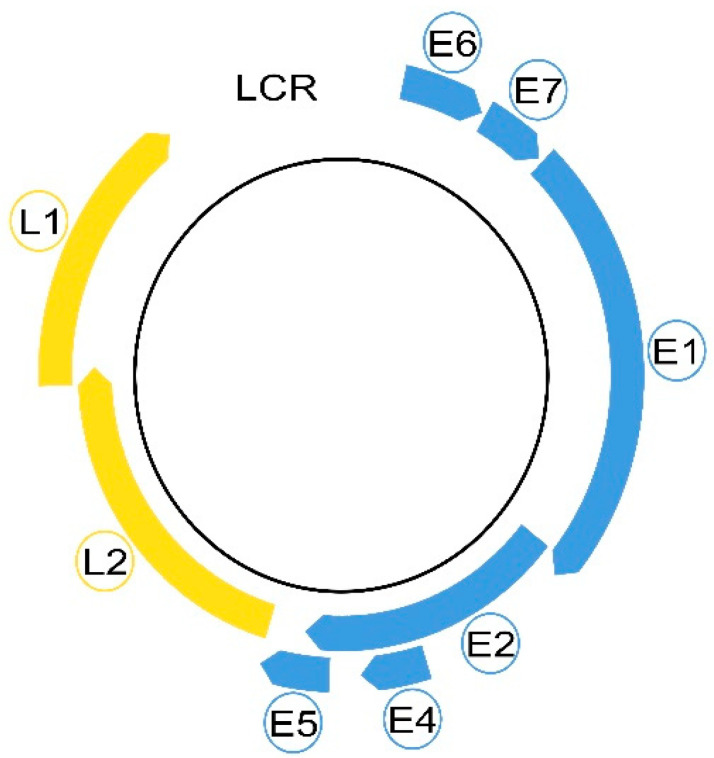
Schematic illustration of a high-risk HPV morphology.

**Figure 2 medicina-58-01103-f002:**
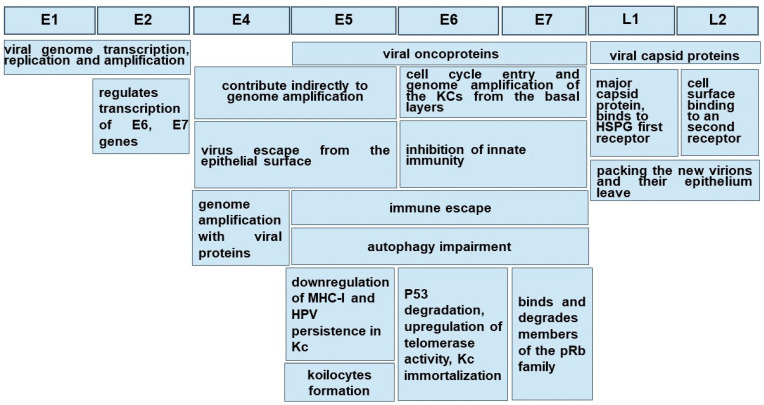
The roles of high-risk HPV main gene products. E1-L2—HPV genes; Kc-Keratinocyte; HSPGs—heparan sulfate proteoglycans.

**Figure 3 medicina-58-01103-f003:**
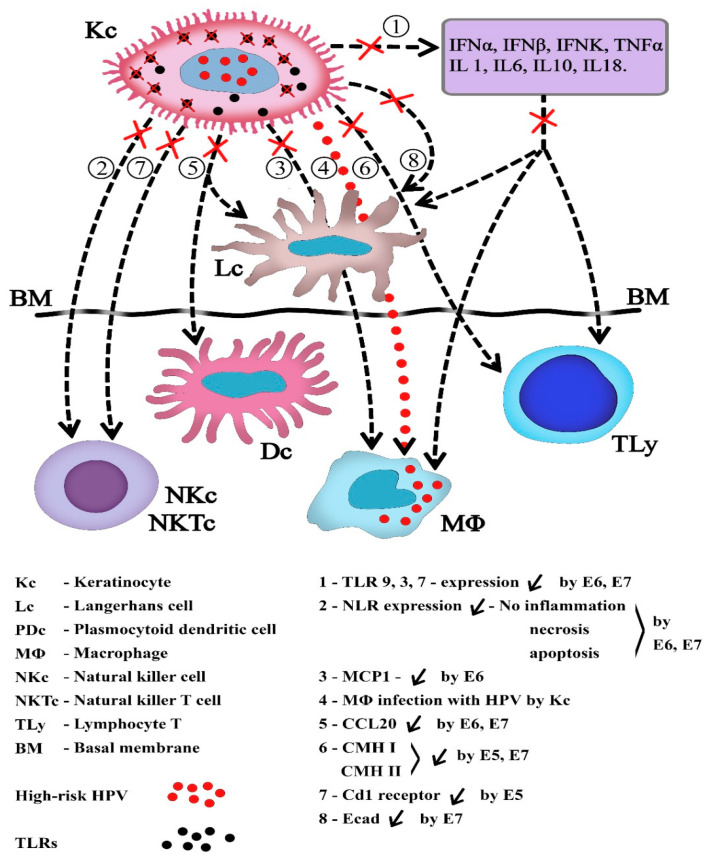
The central role of infected Kcs in suppressing the immune response. (see details in the text below).

**Figure 4 medicina-58-01103-f004:**
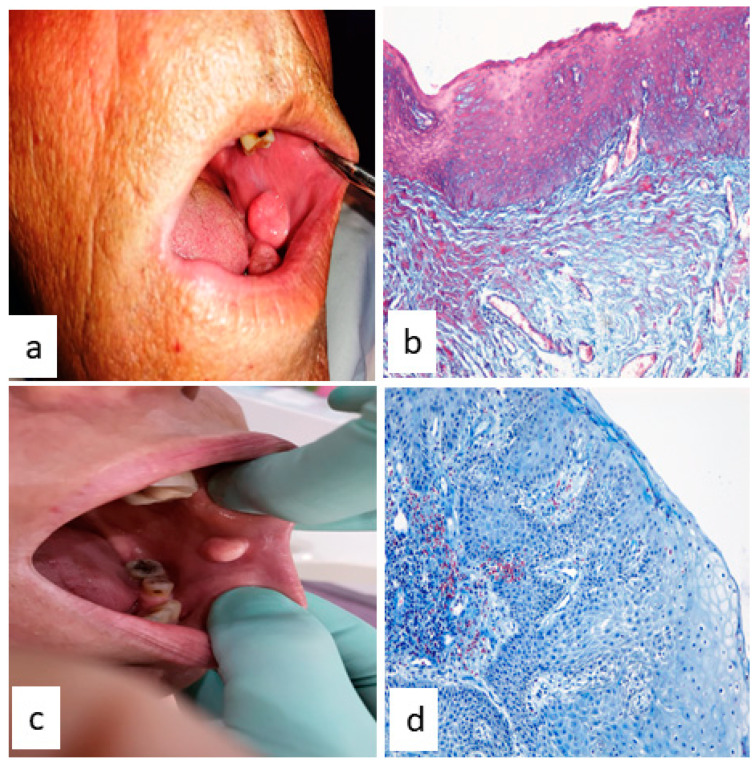
HPV-negative oral papilloma (**a**,**b**) and HPV-positive papilloma (**c**,**d**). (**a**) Macroscopically, tumor lesion located at the jugal mucosa, round shape, size 0.75 mm, pinkish-whitish color, poorly vascularized, non-bleeding; (**b**) Histological aspect of jugal mucosa; acanthosis, parakeratosis and koilocytes presence in the epithelium, (trichrome stain × 200); (**c**) Macroscopically, tumor lesion located at the occlusal line of the jugal mucosa, round-oval shape, size 0.95 mm, red color, richly vascularized, bleeding at the slightest trauma; (**d**) Hiper- and parakeratosis, acanthosis and koilocytes presence in the epithelium; in the basal epithelial layer mitotic figures are present. Proinflammatory cells infiltrate the mucosa (trichrome stain × 400).

## Data Availability

All information discussed in this review is documented by relevant references.

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
