# Peer review of "Oral Papillomatosis: Its Relation with Human Papilloma Virus Infection and Local Immunity—An Update"

_medicina, 2022, doi:10.3390/medicina58081103_

Round 1

Reviewer 1 Report

This Review describes oral papillomas.

There are two types of papillomas, HPV-associated papillomas, and non-HPV-associated papillomas. Still, more details about non-HPV-associated papillomas should be described before writing about HPV-associated papillomas. Including macro and histological pictures of each papilloma in the review would be better. The authors described high-risk HPV in detail, even though low-risk HPV is mainly involved. Table 1 should be revised because the text is difficult to read.

Overall, the review contains many English spelling errors, and the text is not well structured.

We recommend that the text be substantially revised and resubmitted.

Reviewer 2 Report

This article is a comprehensive review of HPV and oral papillomatosis. It is a useful review including the basic and clinical aspects of the disease. However, I think it needs minor revisions.

Table 1 is difficult to read and not clear. I think it should be changed to a diagram that is easier to understand.

Line 119 gens products → gene products

Line 210 ADN → DNA?

Reviewer 3 Report

The manuscript by Andrei et al., collect relevant information in the oral papillomatosis field. The manuscript is well written and presented. Certainly there is a lack of manuscript describing oral papillomatosis focused in local immunity.

I have some comments before considering this manuscript for publication.

Introduction could be shortened, the main focus should be given to local immunity, which is very well organized. Enhancing the figure by adding some details and quality, will provide readers with a magnificent image that surely will be well appreciated.
There are several typos along the text, please check

Probably a table summarizing the site, molecules, effects, cell type and HPV protein/genotype involved will help to focus the attention when comparing data along the different sites.

Round 2

Reviewer 1 Report

The authors responded to my comments. However, the text is still not structured, and the contents are insufficient for the review paper. I recommend that you will rewrite it and resubmit it again.

Minor comment1

Line231, HPV-6 and -11 are responsible for RRP that can convert to cancer because their DNA is maintained in the episomal state within the nucleus similar to high risk HPVs [36, 57].

I could not find the contents mentioned above in references 36 and 57. RRPs are rarely oncogenic, but most are not. Low-risk HPVs exist in the episomal form in the host cell's nucleus. On the other hand, high-risk HPVs are integrated into the host genome. Also, low-risk HPVs E6 and E7 are pretty different from high-risk HPVs and have no or low ability to degrade p53 (Fu et al., 2010) and pRB. For this reason, RRP is rarely carcinogenic. And it has been reported that this rare oncogenesis of RRP may be caused by the integration of the HPV gene into the host cell genome, thereby knocking out the host gene (Huebbers et al., 2013). As such, the above representation is inappropriate.

Fu L, Van Doorslaer K, Chen Z, Ristriani T, Masson M, et al. (2010) Degradation of p53 by Human Alphapapillomavirus E6 Proteins Shows a Stronger Correlation with Phylogeny than Oncogenicity. PLOS ONE 5(9): e12816. https://doi.org/10.1371/journal.pone.0012816

Huebbers CU, Preuss SF, Kolligs J, Vent J, Stenner M, Wieland U, Silling S, Drebber U, Speel EJ, Klussmann JP. Integration of HPV6 and downregulation of AKR1C3 expression mark malignant transformation in a patient with juvenile-onset laryngeal papillomatosis. PLoS One. 2013; 8:e57207.

Minor comment2

Table 1 is still difficult to read.

It is not a good idea to pack long sentences into small boxes. It would help if you tried to make the letters bigger or shorten the sentences.

Author Response

Dear Reviewer,

We would like to thank you again for the time you spent for a thorough analysis of our paper and for your valuable observations.

We analyzed carefully your comments and below you will find a few remarks regarding our work related to them:

  • ”The authors responded to my comments. However, the text is still not structured, and the contents are insufficient for the review paper. I recommend that you will rewrite it and resubmit it again.”

 – Our paper was structured so the data presented are centered on the argument proposed in the title – the role of the local immunity in HPV-induced oral papillomatosis. In our paper, as well as in the literature read, high-risk HPVs are studied more intensively because they are ones that can cause oral carcinomatosis. Therefore, the rest of the data have been presented in this context to delineate the main features regarding HPV-target cells interaction.

  • Minor comment 1

”HPV-6 and -11 are responsible for RRP that can convert to cancer because their DNA is maintained in the episomal state within the nucleus similar to high risk HPVs [36, 57]. I could not find the contents mentioned above in references 36 and 57… As such, the above representation is inappropriate.”

  • Thank you for this observation. Unfortunately, trying to condense the information presented, based on references 36 and 57, the meaning of the phrase was confused. Based on your remark, we read carefully and rephrased the basic idea in the new version sent:

It’s a general agreement that the Kcs of the oral mucosa are the reservoir of HPV, and the most common behavior of the oncogenic HPVs, especially HPV-16 and-18, but also of the low-risk HPV-6 and -11, is the virus persistence [56]. In order to maintain itself inside the host cells, either viral DNA is integrated into the host genome as for high-risk HPVs or it is maintained as episomes that tether to host DNA as for low-risk HPVs. The integration of HPV DNA alters the expression of key genes in the host and promotes genomic instability and oncogenic progression [56].

HPV-6 and -11 are responsible for RRP that rarely can convert to cancer [36]. It has been reported that this rare oncogenesis of RRP may be caused by the integration of the HPV gene into the host cell genome, thereby knocking out the host gene [57].

Research data provide evidence that HPV-11 poses a higher potential of malignant transformation [57]. The oncogenicity differences between these low-risk HPVs could be related to the ability of E6 and E7 proteins of HPV-11 to have a more efficient interaction with host cell ”oncogenic targets”, than those from HPV-6 [36, 58]. It was demonstrated the involvement of E2 protein from high-risk HPV-16/18 and HPV-11 to maintain HPV DNA as “mini-chromosomes” in dividing cells through its direct interaction with the mitotic spindles [56]. Still, HPV-6 and -11 proteins has a different ability to interfere with tumor suppressors p53 and pRb and other cell cycle regulators than those from high‐risk HPVs [36].

  • Minor comment2

”Table 1 is still difficult to read. It is not a good idea to pack long sentences into small boxes. It would help if you tried to make the letters bigger or shorten the sentences”

  • Taking into account your observation, we made the table again with fewer words, changing the letters and the background to be better understandable.

Finally, we hope that these changes will improve the context intended in order to be accepted for publication. Thank you again for your review!

The Authors